
# A miniature Portable Emissions Measurement System (PEMS) for real-driving monitoring of motorcycles

Michal Vojtisek-Lom[1,2], Alessandro A. Zardini[3] , Martin Pechout[2], Lubos Dittrich[2], Fausto Forni[3],
François Montigny[3], Massimo Carriero[3], Barouch Giechaskiel[3], Giorgio Martini[3]

[1]Center for Vehicles for Sustainable Mobility, Faculty of Mechanical Engineering, Czech Technical University of Prague, Technicka 4, 166 07 Prague, Czech Republic
[2]Department of Vehicles and Engines, Faculty of Mechanical Engineering, Technical University of Liberec, Studentska 2, 461 17 Liberec, Czech Republic
[3]European Commission, Joint Research Centre (JRC), Ispra, Italy

*Correspondence to*: M. Vojtisek-Lom (michal.vojtisek@fs.cvut.cz)

**Abstract.**

We present an exploratory study carried out with a new miniature portable emission measurement system (Mini-PEMS) specifically designed at the Technical University of Liberec (CZ) for applications on 2-wheeler vehicles owing to its reduced size (45 x 30 x 20 cm) and weight (≈ 15 kg). It measures the exhaust gas concentrations of hydrocarbons, carbon monoxide and dioxide with non-dispersive infrared method, nitrogen mono- and di-oxides and oxygen using an electrochemical cell. In addition, the instrument acquires the engine speed, the manifold absolute pressure, the inlet and exhaust gas temperature, the geo-localization and vehicle speed. The exhaust mass flow rate is calculated from engine and emission data. The Mini-PEMS was validated on three 2-wheelers (1 moped and 2 motorcycles) against laboratory-grade instrumentation in the Vehicle Emissions Laboratory of the European Commission in terms of measured concentrations, exhaust flow, fuel consumption and mass emission of pollutants. The mean absolute deviations of gas concentrations were 8% for HC, 8% for CO, 13% for $NO_x$ and 2% for $CO_2$ while the mass emissions (which include the exhaust flow determination uncertainty) were 7% for HC, 7% for CO, 9% for $NO_x$ and 5% for $CO_2$. An agreement of 2% was achieved between the fuel consumption measured in the laboratory and calculated by the Mini-PEMS. As an application, the instrument was tested on board of the vehicles during on-road trips. The emissions measured on-road were consistent among repeated runs, with differences between laboratory and on-road tests much larger than those between Mini-PEMS and laboratory. We found similar or larger HC and NOx real-driving emissions, larger CO from motorcycles and smaller for the moped. Considering its size and weight, the Mini-PEMS proved to be an efficient tool for vehicle monitoring, research and development and could be tested for in-service monitoring applications related to carbon monoxide and nitrogen oxides emissions. A tentative approach to characterize particulate mass and particle number was presented and compared to the existing non-volatile particle number protocol.



# 1 Introduction

Exhaust emissions from internal combustion engines remain to be one of the primary source of air pollution in populated areas, and are believed to be responsible in the European Union (EU) for an order of magnitude more premature deaths than traffic accidents (EEA, 2018).

Two wheelers such as mopeds and motorcycles are known to be strong emitters of particulate matter, hydrocarbons and carbon monoxide and their contribution to air pollution can be amplified by their mode of use, mainly urban and at cold engine conditions (e.g., Kumar et al., 2011; Platt et al., 2014 and references therein). In addition, the aged exhaust emissions from 2-stroke mopeds were demonstrated to be active in the production of secondary organic aerosols, so that the term "asymmetric pollution" was introduced to describe the low share of the total vehicle fleet along with the large contribution to

air pollution (Platt et al., 2014). The use of alkylated fuels with only trace content of aromatic compounds (< 1% vol/vol concentration compared to 29−35% vol/vol concentration in standard gasoline) proved to be extremely beneficial in terms of particulate emissions and secondary aerosol formation as demonstrated by Zardini et al. (2014).

Improvements in engines and exhaust after-treatment technology for 2-wheelers and recent legislative development (introduction of Euro 3 emission standards for mopeds in 2014 and Euro 4 for all 2-wheelers in 2016-2018) have

considerably enlarged the distribution width of emissions (Clairotte et al., 2016). As a result, the circulating fleet of 2-wheelers is very heterogeneous in terms of exhaust emissions, including older Euro 2 mopeds (registered in the period 2002-2013) together with the new Euro 4 motorcycles (introduced in 2016). In addition, the presence of tampered vehicles which are modified for better performance by acting on the propulsion unit and/or after treatment system, may further increase the emission distribution width (Zardini et al. 2016a).

Exhaust emissions from vehicles need to be carefully addressed during laboratory type-approval tests, but they generally tend to be higher during real-driving conditions because of various factors such as driving style (maximum speed, accelerations, and gear-shift strategy), ambient conditions (e.g., temperature, humidity, wind, altitude, and road friction), engine and vehicle general conditions (e.g., tyre pressure, additional weight, aging of the after-treatment system) (Weiss et al., 2011). In order to reduce the discrepancy between real-driving emissions (RDE) and type-approval emissions, the

characterization of RDE using portable emissions monitoring systems (PEMS) has been introduced into the United States and EU legislation for heavy duty vehicles and passenger cars (US-EPA, 2005; EC, 2011; EC, 2012; EC, 2017a; EC, 2018).

The EU type-approval legislation has been recently updated for what concerns 2-wheelers, tricycles and quadricycles such as mini cars and quads (EC, 2013; EC, 2014). These vehicles are grouped in the so called L-category family for which two environmental packages were introduced: Euro 4 from 2016 and Euro 5 from 2020. Additional features to be assessed for

post-Euro 5 environmental measures are stipulated in recital 12 of Regulation EU 168/2013 (EC, 2013) and involve the feasibility of particle number measurements (Giechaskiel et al., 2015) and off-cycle emissions, i.e., non-type-approval driving cycles run on a chassis dynamometer. The baseline concept promoted by the European Commission is that an L-





category vehicle should be clean and energy efficient in each point of its possible operation range, in addition to the type-approval driving cycle (Zardini et al., 2016b).

Before the introduction of Euro 5 for the L-category, the European Commission undertook an environmental effect study (Zardini et al., 2016b; EC, 2017b) which included a preliminary assessment of RDE. The effect study concluded that the

concentrations of pollutants can be measured with a marketable, small, portable system exhibiting an average discrepancy of about 10% during chassis dynamometer tests, except for hydrocarbons affected by more than 50% discrepancy. The disagreement in hydrocarbons measurements stems from different applied techniques: flame ionization detector with the use of a hydrogen cylinder in the test-cell; non-dispersive infrared in the PEMS (in order to exclude safety issues related to the transportation of a hydrogen cylinder on a 2-wheeler). Mass emissions during lab tests and RDE on-road tests compared to

lab tests were instead vehicle and compound dependent, and affected by larger uncertainty and discrepancy due to the additional estimate of the exhaust flow. Exhaust mass flow meters mounted on passenger cars are still too heavy and energy consuming to be deployed on 2-wheelers during RDE tests (see EC, 2017b, for further details) and exhaust pulsations might be an issue for them.

PEMS use analyzers to continuously measure the concentrations of the pollutants of interest, and methods to measure or

estimate the instantaneous flow of exhaust. The data are then synchronized and concentrations multiplied by the corresponding exhaust flow to obtain instantaneous mass emissions of the measured pollutants. Early PEMS relied on simple automotive exhaust analyzers such as those used in the California Air Resources Board periodic inspection programs, with the exhaust flow neglected (Kelly and Groblicki, 1993), inferred from engine operating data (Vojtisek-Lom, 1997), or measured by a Pitot tube (Breton, 2000), or similar device.

With few exceptions (e.g., Lenaers, 2003), the majority of PEMS use standard non-dispersive infra-red (NDIR) analyzers to monitor carbon monoxide (CO) and dioxide ($CO_2$), chemiluminescence or non-dispersive ultra-violet spectroscopy for nitrogen monoxide (NO) and dioxide ($NO_2$), and flame ionization detection (FID) for hydrocarbons (HC). Some early PEMS (Vojtisek-Lom and Cobb, 1997; Breton, 2000) were based on analytical components from garage-grade analyzers (HC, CO and $CO_2$ measured with NDIR, NO and $O_2$ with electrochemical cells). Despite the considerable miniaturization and

improvements in recent PEMS technology, only a couple of PEMS models are at present marketable featuring technical specifications suitable for the L-category: light weight, small size, simultaneous measurements of multi-gas concentrations, engine parameters and geo-localization. The systems designed for passenger cars and heavy duty vehicles weigh tens of kg (about 100 kg including exhaust flow meter and batteries, thus disproportionally increasing emissions when on-board of an L-category vehicle) and are excessively bulky for safe installation on a 2-wheeler.

To our knowledge, the peer-reviewed scientific literature about tailpipe emissions from L-category vehicles relies entirely on laboratory measurements using chassis dynamometers. In some cases, the speed profile was recorded during real-world driving and used for subsequent emissions measurement in a laboratory (Zamboni et al., 2011; Murena et al., 2019).





In this work, a miniature PEMS device (Mini-PEMS, hereafter) suitable to be fitted on motorcycles, scooters and mopeds is introduced and validated against standard laboratory instrumentation. As an application, the Mini-PEMS on-board of three 2-wheelers, 1 moped and 2 motorcycles, was tested on-road and RDE were estimated.

# 2. Experimental

Tailpipe exhaust emissions from 3 L-category vehicles (1 moped, 2 motorcycles) were characterized during i) legislative driving cycles on a roller bench in the Vehicle Emissions Laboratory of the European Commission − Joint research Centre (JRC) and ii) real-driving tests with a miniature on-board measurement system (Mini-PEMS, Technical University of Liberec, Czech Republic). A schematic summary of the deployed techniques in the test cell and in the Mini-PEMS is presented in Table S1. Overall, 8 laboratory emission tests and 9 on-road trips were performed.

## 2.1 Test cell

The test facility has already been described in Zardini et al. (2014, and references therein). Here briefly, the vehicles were driven on a 48'' roller bench (Zoellner GmbH) and their raw and diluted exhaust analyzed in accordance with Regulation EU 134/2014 (EC, 2014); see schematic in Figure 1. Total hydrocarbons (THC), carbon monoxide (CO) and nitrogen oxides ($NO_x$) are regulated by the EU legislation on 2-wheelers aiming at reducing air pollution. Carbon dioxide ($CO_2$) is instead

reported for energy efficiency assessment and global warming related issues, but not yet regulated for this class of vehicles. We monitored the raw exhaust at 1 Hz via a 190 $^{\circ}$C heated line with the following techniques (integrated in the AMA i60 Exhaust Measurement System, AVL): flame ionization detector for THC, chemiluminescence detector for NOx, non-dispersive infrared for CO and $CO_2$. The same on-line techniques were deployed at the constant volume sampler (CVS) after dilution of the exhaust through a Venturi nozzle dilution tunnel (CVS flow ≈ 5 $m^3$/min; average dilution ≈ 25). Finally, a

constant volume sample taken from the CVS flow was collected in tedlar bags and analyzed off-line after the test. The results of the bag analysis yield the legislative emission factors (EFs) given in mass/distance unit. Following a well-established methodology, bag values were dynamically corrected for exhaust extracted from the tailpipe due to raw exhaust sampling by multiple devices. For instance, the range of mass extracted/total mass of $CO_2$ is between 9% for Vehicle 2 and 21% for the moped. In particular, the laboratory setup can provide three sets of EFs obtained from: (i) the raw exhaust

concentrations and flow rate, (ii) the diluted exhaust concentrations and constant flow rate and (ii) the off-line analysis of bag sampling as prescribed by the EU legislation. In the laboratory, two methods are typically applied for the determination of the exhaust flow: (i) the difference between the CVS constant flow and the dilution air flow and (ii) the well-established $CO_2$ tracer method based on raw (tailpipe) and diluted (CVS) $CO_2$ simultaneous measurements (Wiers et al. 1972). Due to the low exhaust flow rate of 2-wheelers, the difference between two flows of similar magnitude (CVS and dilution air)





measured by different flowmeters results in large uncertainty. Therefore, the $CO_2$ tracer is the method of choice in the present work.

The AVL Particle Counter (APC) 489 (AVL, Graz, Austria), compliant with the light-duty vehicle regulations, was connected to the dilution tunnel to measure non-volatile particles (Giechaskiel et al., 2010). The Volatile Particle Remover

(VPR) of the measurement system consisted of a hot dilution diluter at 150°C, an evaporation tube at 350°C, and a final dilution in a porous diluter with room-temperature filtered air. Downstream the VPR a butanol Condensation Particle Counter (CPC) (model TSI 3790) with 50% counting efficiency at 23 nm (d50% = 23 nm) measured solid particles (Giechaskiel et al., 2009). Non-volatile particle number measurements at the tailpipe were performed using a Nanomet 1 system. This consists of an MD19-2E rotating disc diluter (Hueglin et al., 1997) followed by an ASET15-1 thermodiluter.

The sample is diluted at the sample point with the rotating disc diluter using conditioned air at 150°C. The diluted sample is then thermally treated at 350°C in an evaporation tube and subsequently diluted in a simple air mixer diluter at a rate of 10:1. A PCRF of 150 was employed in these tests (primary dilution of 15). A TSI 3790 CPC having a 50% cut-off size at 23 nm was connected.

## 2.2 Vehicles and driving cycles

The technical specifications of the vehicles under investigation are given in Table 1. We have chosen 3 popular in-use 2-wheelers type-approved and sold in the European market based on the stocktaking and market sales analysis in Clairotte et al. (2016): 1 moped and 2 medium performance motorcycles, as per EU terminology (EC, 2013). The EU legislation sets the emission limit values for the gaseous pollutants THC, $NO_x$ and CO depending on the Euro standard of homologation and the vehicle L-subcategory, as summarized in Table S2. Note that the provisions for the durability of after-treatment devices, i.e.

compliance with limit values within a prescribed mileage, were not enforced for Euro 2 and Euro 3 L-category vehicles. Therefore it could be possible not to comply with emission limits at the time of testing. The 2-wheelers were fuelled with E5 reference petrol containing ≈ 5% vol/vol of ethanol (see EC, 2014, for reference fuel specifications) and their fuel consumption measured with a fuel consumption measurement system (KMA, AVL). Figure 2 and Table 2 describe the two driving cycles for tailpipe emission monitoring: the United Nations ECE-R47 (UNECE, 1981) driving cycle run by the

moped and the Worldwide-harmonized Motorcycle Test Cycle (WMTC) for motorcycles (EC, 2014). The ECE-R47 test cycle consists of eight elementary, consecutive cycles and lasts 896 s in total. Only during Phase 2, also called hot phase (the latter 4 elementary cycles), the exhaust gas should be sampled to produce the EFs described above for type-approval purposes of Euro 2 mopeds (as of July 2014, the type-approval of vehicles requires the sampling of the entire cycle - Euro 3 test procedure for mopeds). However, we sampled and reported both cold and hot phases of the ECE-R47 cycle in order to

maximize the range of application of the Mini-PEMS and to report more realistic EFs. The WMTC cycle is mandatory from the introduction of the Euro 4 step for motorcycles, while it was an alternative option for type-approval of Euro 3 motorcycles as those in our study. The prescribed WMTC for a specific vehicle is assigned depending on engine



displacement and vehicle maximum speed out of a set of 5 different driving cycle types (EC, 2014): Vehicles 2 and 3 performed the WMTC type 2-2 and type 2-1, respectively (see Figure 2). The moped was additionally emission-tested over a wide-open throttle cycle consisting of running the vehicles at constant roller speeds with the throttle plate fully open in order to increase the dynamic range of Mini-PEMS measurements.

The real-driving tests were performed for each vehicle by repeating 3 times a 2.24 km nearly flat-land loop (average altitude ≈ 200 m above sea level; altitude range ≈ 10 m) mimicking urban/suburban driving with cold or hot engine start; see details in Table 2 and Figure 3. The loops were chosen purely for the Mini-PEMS validation and were neither designed to be comparable to legislative driving cycles on the roller bench, nor to fulfil PEMS legislation requirements in EU, which at present are specific only to passenger cars and heavy duty vehicles.

## 10   2.3 Mini-PEMS

The on-road tests were conducted with a miniature portable on-board emissions monitoring system (Mini-PEMS), according to a general concept developed by Vojtisek-Lom et al. (1997; 2009), where instantaneous emissions are determined as a product of the measured concentrations of the pollutants of interest and the estimated mass flow rate of the exhaust. The parameters of the system are given in Table 3, and a schematic in Figure 4. The system samples undiluted raw exhaust at

nominally 8 litres per minute. The sample is cooled down by natural convection in the sample line, passed through a condensation bowl from which the condensate is continuously removed by a pump, and reheated in a copper tube immersed in a bath of transformer oil heated by resistance heating to approximately 60 ºC. The sample is then divided into two paths. In one path, the sample is filtered, passed through a non-dispersive infra-red (NDIR) analyzer to measure the concentrations of HC, CO and $CO_2$, and subsequently divided into three electrochemical cells measuring the concentration of oxygen,

nitrogen monoxide (NO) and nitrogen dioxide ($NO_2$). In the second path the sample runs through a semi-condensing integrating nephelometer (635 nm, 45º forward scattering) tuned to provide reading proportional to particle mass concentrations, and subsequently through a heated ionization chamber ($^{241}$Am, 30 kBq), where the particles present in the chamber deplete the ions, decreasing the current between two electrodes on which a small voltage differential is applied. The change in ionization current has been shown to correlate to the total particle length, with a useful measurement range starting

at levels corresponding to particle number concentrations of approximately $5 \cdot 10^5$ particles/$cm^3$ (Vojtisek-Lom, 2011). These methods, although not measuring directly mass or number, can be correlated well to the current methods, i.e. gravimetry of filters and condensation particle counters, respectively (Giechaskiel et al. 2014).

The engine speed was measured by an optical sensor pointed to a reflective tape placed on the engine cooling fan mounted on the crankshaft. Alternatively, the setup can determine the engine speed by sensing signals from spark plug or injector

wiring, or using vibration sensors. Access to the intake manifold was obtained either from the existing vacuum system, or, in its absence, by drilling a small hole into the intake manifold downstream of the throttle plate; see Picture S1.



Vehicle speed and position were measured by a Global Positioning System (GPS) receiver. All data were acquired by a built-in industrial computer. The system has a footprint of 45 x 31 cm, a height of 18 cm, and mass of 13 kg, and operates on 9-14 V with a consumption of approximately 50 W. The total mass of the system, including power electronics, sensors, cables and sample lines, was 17.7 kg. The setup was mounted on the luggage rack of the 2-wheelers and powered by an external battery

(12V, 20Ah LiFeYPo, 3.4 kg, allowing 3-4 hours of autonomy) attached to the platform for the rider's feet. The installation of the system is shown in Picture S2, with details of engine speed sensor, intake manifold pressure sampling point, and exhaust sampling given in Picture S1. A second Mini-PEMS system (Mini-PEMS No. 2) based on the same measurement techniques explained above, but smaller and lighter than Mini-PEMS No. 1, was deployed in parallel during the 3 roller bench tests of Vehicle 2; see Picture S3. It featured a built-in battery allowing 3-4 hours of autonomous run time, a footprint

of 40x20 cm, height of 20 cm, and total mass of less than 10 kg.

The fuel consumption measured by the KMA flow meter was compared to the fuel consumption inferred from the emissions measured by the Mini-PEMS and by the laboratory. In both cases, the mass emissions of carbon were calculated from THC, CO and $CO_2$ emissions and then divided by the 86.6% carbon content of the fuel to obtain fuel mass flow rate (MFR):

$$MFR \left[\frac{g}{s}\right] = \frac{M_C/_{M_{CO}} * CO\left[g/_s\right] + M_C/_{M_{CO_2}} * CO_2\left[g/_s\right] + M_C/_{M_{HC}} * HC\left[g/_s\right]}{0.866}$$

(1)

where $M_x$ is the molar mass of carbon (C), CO, $CO_2$ and HC, $x$ [g/s] is the mass flow of compound $x$, 0.866 represents the mass fraction of carbon in the fuel. A molar weight of HC = 14 g/mol was calculated from fuel properties. Emissions over a test run are obtained by integrating the instantaneous emissions, and dividing the total emissions (in grams per test) by the total distance driven to obtain emission factors in g/km. Note that the EU legislation prescribes fuel consumption calculations based on integrated off-line bag results, partly to avoid the uncertainty caused by synchronization and dynamic

correction of instantaneous data.

The exhaust flow rate is inferred from the intake air flow, which is estimated using the speed-density method from measured engine speed and intake manifold pressure and temperature, engine displacement and known or assumed engine volumetric efficiency (Vojtisek-Lom, 1997; Vojtisek-Lom and Cobb, 1998). The original formula assumed general volumetric efficiency (0.9-0.95 at higher loads) of automotive engines, which is excessively high for small engines. Empirical

adjustments were therefore introduced based on laboratory tests to account for considerable lower volumetric efficiencies for small motorcycle engines at smaller engine speeds. The intake flow rate can therefore be obtained from:

$$m_{air} \left[g/_s\right] = 0.0289 \left[kg/_{mol}\right] * \eta_{vol} * rpm\left[min^{-1}\right] * MAP \frac{[kPa]}{T_{int}[K]}$$

(2)





where $m_{air}$ is the intake mass flow rate, 0.0289 is the molar weight of ambient air in the standard atmosphere, $\eta_{vol}$ is the engine volumetric efficiency, $rpm$ is the engine rotational speed, $MAP$ is the manifold absolute pressure, $T_{int}$ is the intake air temperature.

Two scenarios were considered:

a) constant volumetric efficiency of $\eta_{vol\text{-}const} = 0.5$ for all combinations of engine speed and load; this approach is believed to provide a reasonable estimate for naturally aspirated engines without EGR (see Section 3)

b) engine speed- and load-dependent volumetric efficiency $\eta_{vol\text{-}var}$ calculated using the empirical formula derived by comparing instantaneous fuel consumption obtained by direct measurement and calculated from Mini-PEMS data:

$$\eta_{vol-var} = \left[ \left( \eta_{vol-const} + \frac{1}{CR} \right) * \left( 1 - \frac{\frac{p_{bar}}{CR}}{MAP} \right) \right] + \left[ \frac{rpm - 5000}{32000} \right] - \left\{ 0.0015 * \left[ p_{bar} * \left( 1 - \frac{1}{CR} \right) - MAP \right] \right\}$$

(3)

where $CR$ is the dimensionless engine compression ratio and $p_{bar}$ is the barometric pressure in kPa.

In both scenarios, the empirical values were obtained iteratively by comparing instantaneous mass fuel flow rate calculated by the Mini-PEMS with the instantaneous fuel consumption measured by the laboratory (results not reported here).

While exhaust flow can be inferred from known exhaust and fuel composition and from either fuel or intake air flow

(Vojtisek-Lom and Cobb, 1997), for simplicity, the intake flow as described above was used in lieu of the exhaust flow in Mini-PEMS fuel flow and exhaust calculations. Additionally, the volumetric efficiency included a variable dry-to-wet correction (Giechaskiel et al. 2019a) applied to measured $CO_2$ and CO.

The emission factors from PEMS measurements where finally calculated as follows:

$$X \left[ \frac{g}{s} \right] = m_{air} \left[ \frac{mol}{s} \right] * [X] * M_x \left[ \frac{g}{mol} \right]$$

(4)

where $m_{air}$ is the molar flow, $[X]$ is the volume fraction of the $X$ compound in the undiluted exhaust, and $M_x$ is the molar weight of the $X$ compound.

The validation of the Mini-PEMS against laboratory instrumentation over roller bench tests performed in accordance with Regulation 134/2014 (EC, 2014) was based on the comparison of (i) the pollutant concentration profiles from the raw exhaust, (ii) the exhaust flow rates and (iii) the emission factors. Ultimately, EFs obtained on-road with the Mini-PEMS

should be compared with (i) laboratory EFs originated from the legislative method to assess the severity of the test cycle with respect to real-driving, and/or with (ii) emission limit values prescribed by the legislation in order to validate the environmental performance of the vehicle, as prescribed in the recent Euro 6 legislation dealing with in-service conformity of passenger cars (EC, 2018; Varella et al., 2018). As expected from basic principles governing the combustion of small gasoline engines, $NO_2$ emissions were negligible (up to 1% of NO) and therefore not reported.



# 3 Results and discussion

## 3.1 Mini-PEMS versus laboratory

### 3.1.1 Pollutant concentrations

Table 4 compares gaseous average concentrations of THC, CO, NOx and $CO_2$ obtained with laboratory instrumentation
(bench) against those from the two Mini-PEMS during 8 roller bench tests according to the methods described in section 2.
Single test percentage deviations, mean absolute percentage deviations (MAPD) for each vehicle and overall for the fleet as
well as maximum and minimum deviations are included. Considering Mini-PEMS No. 1, the overall MAPD was 2% for
$CO_2$, 8% for HC and CO, and 13% for NOx, indicating very good agreement with laboratory instruments. The best
agreement was obtained for $CO_2$, which typically exhibits the largest and most stable concentration profile during emission
testing from internal combustion engines. The worst single-test deviation was for NOx in Test 1 (24%) due to substantial
zero drift of the Mini-PEMS sensor.

An example of raw exhaust concentration profiles during Test 5 and associated 1-to-1 plots with linear regression
coefficients are plotted in Figures S1 and S2, respectively. The two Mini-PEMS well followed the dynamic emission pattern
down to a few seconds resolution. The largest discrepancies were related to the signal peaks where the longer or different
response time of the Mini-PEMS results in peak magnitude underestimation and subtle shifts in instantaneous values during
transients, which negatively affects the coefficients of determination ($R^2$) in Figure S2.

Table S3 summarizes the comparison between the two Mini-PEMS and the bench in terms of coefficients of determination.
The results vary substantially among the tests. Differences in instantaneous readings during transients and peaks due to
limited 1 Hz resolution and different time response constants of the respective instruments, possible dilution of the exhaust
sample at idle due to comparable amounts of available exhaust and sampling lines (about 20 lpm for the moped), and in case
of HC, deposition of semi-volatile hydrocarbons (known as hydrocarbon hangup) in the Mini-PEMS sampling train, were
identified as the main responsible to lower values of coefficients of determination.

In order to assess the performance of the two Mini-PEMS during the cold phase of the test cycles when the temperature of
the sampled exhaust increases from about 25 ℃ up to 200-300 ℃ in about 3 minutes, the approach above was repeated for
separate cold and hot cycle phases as shown in Table S4. Looking at cold start Tests 1, 5, 6, and 8, the sampling of the cold
phase did not represent a critical issue for the two Mini-PEMS as the performance is comparable or better than in the hot
phase. However, the phase split highlights a large discrepancy between laboratory and Mini-PEMS No. 1 in Test 8. The
average HC concentrations measured by Mini-PEMS No. 1 during Test 8 were, compared to the laboratory, lower during the
first phase of the test, and higher during the second phase. This demonstrates a shortcoming of the unheated sampling and
measurement system used in the portable system, in which semivolatile organic species (i.e., fuel and oil vapors) present in
high concentrations during a cold start partially condense within the system, causing the readings to be initially lower, and
later are re-entrained into the sampling stream, causing readings to be higher. This phenomenon occurs initially in the





exhaust system, but it is partially dampened by the heated gas line of the FID used as a reference. Concerning Mini-PEMS No. 2, the mean concentrations reported in Table 4 show excellent $CO_2$ correlation with the laboratory (MAPD < 1%) and very good agreement in terms of CO and NOx (MAPD = 7% and 13%, respectively). This good performance, along with similar results from Mini-PEMS No. 1, confirmed the reliability of the setup design when measuring pollutants concentration

in vehicular exhaust. The larger deviation for HC (MAPD = 20%, with largest single test deviation of 32%) can be explained with the condensation of semivolatile compounds in the sampling system (hydrocarbon hang-up) and different sensitivity of the NDIR sensors used in the 2 Mini-PEMS to individual hydrocarbons.

### 3.1.2 Exhaust flow rates

The determination of the exhaust flow is the second key parameter involved in the calculation of mass/distance emission

factors and fuel consumption from instantaneous data, beyond the chemical concentration discussed above. The legislative emission factors calculations do not rely on the exhaust flow, while those from Mini-PEMS do (see Chapter 2). The laboratory exhaust flow calculated with the $CO_2$ tracer method should not be considered as a legislative reference against which to validate the two Mini-PEMS, but as the state of the art estimate available. The upper panel of Figure 5 shows the comparison between the exhaust flows calculated from laboratory instrumentation and from the two Mini-PEMS for Test 5.

The agreement is qualitatively good considering that the two approaches were completely different (see section 2 for details). The largest deviations occurred during idle (very low flow rate ≈ 30 L/min) and during quick deceleration phases when the abrupt change in $CO_2$ concentrations causes narrow peaks in the tracer method curve. On average for all tests, the coefficients of determination between the 2 methods are fairly good ($R^2 ≈ 0.7$) as reported in Table S3, confirming that (i) the Mini-PEMS applies an approach which gives similar results to the $CO_2$ tracer method commonly used when sampling

directly from the tailpipe and that (ii) the exhaust flow remains the critical parameter to assess when performing raw exhaust measurements. The validation of the $CO_2$ tracer method is discussed below in terms of emission factors. Note that at the moment there are no exhaust flow meters to be deployed on 2-wheelers on-road. Future technical developments may fulfil this gap; however, exhaust flows from 2-wheelers are intrinsically difficult to measure precisely due to small flow rates (especially at low engine loads) and, in the presence of one cylinder, to flow pulsations.

### 3.1.3 Fuel consumption

Fuel efficiency of vehicles is officially reported in EU type-approval certificates and calculated from integrated emissions over the entire legislative emission tests (test type VII, EC, 2014). Neither instantaneous fuel consumption during roller bench tests nor real-driving fuel consumption (both calculated from instantaneous emissions, see section 2.3) are required. Nevertheless, great attention was recently paid over real driving fuel consumption data (see e.g., Fontaras et al., 2017, and

references therein) especially when results differ from the official values reported by manufacturers.





The fuel consumption calculated with the carbon balance method from the Mini-PEMS and laboratory HC, CO and $CO_2$ emissions is compared with the fuel consumption measured by the fuel flow meter (reference) in Figure 5. The agreement was good except for the underestimated relative maxima of laboratory results compared to the fuel flow meter. Table 5 compares fuel consumption results from 3 tests for (i) the fuel flow meter, both in mass and volume of fuel (with

instantaneous control of fuel density and temperature), (ii) the laboratory instrumentation (legislative results from bag analysis of diluted emissions and integrated instantaneous results from raw emissions), (iii) the two Mini-PEMS. In all cases the deviation from the reference was ≤ 10%, with an excellent performance of the two Mini-PEMS (deviation < 5%). This indicates that:

- The legislative method based on carbon balance and diluted emissions is a good approximation to determine fuel
consumption during roller bench tests;
- The same method applied to raw exhaust sampling is in good agreement with the reference;
- The Mini-PEMS is a valuable instrument to assess the fuel consumption and can provide useful insights during real-driving tests. Error attributed to using assumed rather than actual volumetric efficiency contributes substantially to the differences between PEMS and laboratory data and can be reduced by experimental determination of $\eta$ in the
laboratory from emissions or fuel consumption data.

### 3.1.4 Mass emissions

As an example, mass flow rates of HC, CO, NOx and $CO_2$, during Test 5 from the two Mini-PEMS are compared against the laboratory instrumentation in Figure 6 assuming variable (engine speed and load dependent) volumetric efficiency (see section 2.3). The agreement is qualitatively very good considering that mass flows are affected by combined uncertainties on

the chemical concentration and on the exhaust flows previously discussed. Clearly, the Mini-PEMS could be used to monitor dynamic exhaust mass flows and for instance to spot tampered vehicles (see Zardini et al, 2016a) or to investigate on-road after-treatment strategies for pollution reduction. Table 6 summarizes the emission factors for laboratory tests performed on the legislative test cycles ECE-R47 and WMTC assuming variable volumetric efficiency. WOT Tests 3 and 4 were excluded from the analysis to avoid reporting artificially high EFs. Similarly to the assessment of concentration profiles, EFs were in

very good agreement: the overall MAPD for Mini-PEMS No. 1 was below 10% for all compounds and down to 5% for $CO_2$. The worst single-test deviation was -22% for NOx in Test 2. Mini-PEMS No. 2 exhibited a generally lower performance in terms of deviations due to lower sample flow during the tests and to the smaller specific data set. Nevertheless, a MAPD < 20% for all compounds indicated good agreement between instruments.

The EFs of HC, CO, NOx and $CO_2$ for each cycle as calculated by the two Mini-PEMS and as measured by the laboratory

are compared in Figure 7 for a total of 6 tests. Three sets of EFs from Mini-PEMS were compared against the laboratory: EFs from Mini-PEMS No. 1 were obtained using both variable (speed- and load-dependent) and constant engine volumetric efficiency. For Mini-PEMS No. 2 the comparison is plotted only for variable volumetric efficiency in order to retain clarity.



Individual data points represent the emissions factors for each test, with laboratory and Mini-PEMS data on the horizontal and vertical axes, respectively. Linear regression is shown for Mini-PEMS No. 1 variable volumetric efficiency. It is apparent that except for HC, there is little discrepancy between EFs calculated by Mini-PEMS No. 1 and No. 2. The calculations using constant volumetric efficiency in Table S5 show a higher, but still relatively small (overall average < 10%), deviation from the laboratory data.

In our analysis, the EFs from the legislative method were chosen as a reference against which the Mini-PEMS was compared in line with current real driving emission legislation and in order to assess the performance of the instrument in typical cases when only the diluted exhaust sampling is present in the laboratory (e.g., type-approval). For sake of completeness, mass emissions were also calculated from the raw exhaust with laboratory instrumentation and further comparison between EFs from Mini-PEMS, EFs from bag sampling and EFs from the raw exhaust sampled with bench instrumentation was carried out. Figure S3 displays the EFs with the three methods broken down by driving cycle phase. The bag and raw exhaust methods with laboratory instrumentation were in good agreement (NOx MAPD = 16%; THC MAPD = 13%, averaged on the cold and hot cycle phases of all vehicles) or very good agreement (MAPD = 8% for CO and 4% for $CO_2$), in line with the historical records of the JRC laboratory. The Mini-PEMS performed well against the raw exhaust method, slightly worse than against the legislative method discussed above due propagation of uncertainties on the exhaust flow. NOx MAPD per phase, shown in Tables S6 and S7 for variable and fixed volumetric efficiency, remained unchanged, but averaged MAPD for HC, CO and $CO_2$ increased up to 20%, 15%, and 10%, respectively.

In order to contextualize our work in the current EU real-driving legislative framework laid down in Regulation EU 2017/1151 (EC, 2017a) and Regulation EU 2018/1832 (EC, 2018) applicable to passenger cars, the Mini-PEMS derived EFs were compared to the legislative EFs in Figure 8. For each pollutant apart from THC (not included in the EU RDE legislation), two graphs are plotted: for each phase of the test separately on the left, and for the whole test on the right. The compound specific tolerances introduced by Regulation EU 1151/2017 (EC, 2017a) to account for the additional measurement uncertainty of PEMS relative to standard laboratory equipment are plotted as dashed red lines. It is apparent from Figure 8 that in most cases, the differences between the PEMS and the laboratory fall within the margins of uncertainty applicable to type-approval grade PEMS, and outer values are relatively close to the margins. The tests outside the margins are mostly of Vehicle 1 (50 cc engine), confirming the experimental challenge posed by measuring emissions from small vehicles.

The particulate matter measured with the mini-PEMS was compared with the number of non-volatile particles from the tailpipe and the dilution tunnel (Giechaskiel et al. 2019b), as applicable to recent diesel and direct injection passenger cars and heavy duty vehicles (Giechaskiel et al. 2018). Note that the mini-PEMS measures particles by light scattering and by an ionization chamber in a sample of raw exhaust, without removal of the semi-volatile compounds. In Table S8, the non-volatile particles sampled at the dilution tunnel and at the tailpipe are compared to the total number (including volatiles) inferred from ionization chamber measurements and particulate mass emissions inferred from laser scattering (based on



calculated exhaust flow and concentrations measured in the raw exhaust). For the two R47 tests listed in Table S8, the comparison of second-by-second particle number emissions (particles per second) is plotted in Figure S4.

As expected from small, port-fuel injection, 4-stroke gasoline engines, PM emissions are low (< 1 mg/km compared to the limit of 4.5 mg/km (Giechaskiel et al. 2019c) and total particle number emissions measured by the Mini-PEMS is in the

range from similar up to double relative to non-volatile particle number emissions > 10 nm (Giechaskiel et al. 2015). The specific vehicle had emissions on the same order of magnitude as the limit value ($6 \cdot 10^{11}$ particles/km) applicable to recent passenger cars.

### 3.1.5 Mini-PEMS No. 1 vs Mini-PEMS No. 2

The accuracy of the instantaneous mass emissions rates is dependent on the quality of the concentration data, exhaust flow

data, and synchronization between the two data streams (Giechaskiel et al. 2018b; Vojtisek-Lom et al. 2018). The principal uncertainty associated with the exhaust flow data is the estimation of the engine volumetric efficiency. The uncertainty associated with the measurement of engine speed and intake air pressure and temperature is rather small, the sensor response relatively fast, and uncertainty of the sensor outputs relatively small. The uncertainties associated with the concentration data are comparable between the two PEMS instruments, as they use the same detection principles and similar components. The

synchronization of the data is comparable between the instruments. The difference between the two Mini-PEMS, as apparent from Table 4, Table 6, Figure 5 and Figure 6, is therefore closer to the unit-to-unit variance than to a difference between two different technologies. Both the concentration and the mass emissions data presented here suggest that the difference between the two Mini-PEMS units is relatively small (up to 0.04 g/km HC, 0.01-0.07 g/km CO, < 0.01 g/km $NO_x$) and slightly smaller or comparable to the differences between Mini-PEMS and laboratory (up to 0.03 g/km HC, 0.06-0.17 g/km

CO, < 0.02 g/km $NO_x$), suggesting that systematic errors (such as bias caused by using NDIR technology for HC detection, or by estimating engine volumetric efficiency) represent a substantial part of the overall uncertainty of the Mini-PEMS measurement.

### 3.2 Real-driving emissions

The PEMS emission factors during on-road trips are typically compared to those obtained in the test cell in order to identify

discrepancies between real driving and laboratory emissions (e.g., during in-service conformity checks). We applied the same procedure with few notes of caution. Two-wheelers have so far very limited on-board diagnostics: the introduction of emission monitoring OBD is planned in 2020. The presence of a defeat device to reduce the emissions during type-approval test cycles can be therefore excluded a-priori, i.e., large discrepancies between roller bench and on-road tests due to artificial modification of the engine map were not expected. Secondly, the real-driving emission test is not included in the current

legislation for 2-wheelers and there are no comprehensive studies from which to derive the technical definition of an agreed typical on-road trip for 2-wheelers with associated boundary conditions and tolerances. Agreement between contracting



parties was reached during the definition of the legislative laboratory WMTC of subcategory L3e-A3 (high performance motorcycles) which was then applied and adapted to the other members of the L-category family (EMISIA, 2013). Finally, mopeds (subcategory L1e-B) are subject to legislative limitation of maximum speed (typically 45 km/h, 25 km/h in some EU countries) which corresponds to their maximum WMTC speed: their laboratory emission testing covers the entire speed

range. Our on-road trips are neither based on the average driving, nor contain a prescribed mix of low/middle/high speed, nor respect validity criteria; they are simple tools to demonstrate the capabilities of the Mini-PEMS. Vehicle speed and positive acceleration of on-road trips as a function of the actual distance relative to the start were given in Figure 3.

Exhaust concentration and real driving emission factors during on-road trips of the three vehicles are reported in Table 7. By comparing on-road against laboratory data in Table 4 and Table 6, it is apparent that the mean concentrations of HC for

Vehicle 1 on the road (3603-4037 ppm) are comparable to the WOT test (2089-3722 ppm) and higher than the R47 test (1455-1550 ppm) in the laboratory. Similarly, the concentration of HC was doubled during the on-road cold start test (2089 ppm) than in the laboratory cold start WMTC (1197 ppm) for Vehicle 3. The concentrations of CO on the road were lower for Vehicle 1 (0.5-0,9%) than in the laboratory (0.8-1.4%) and higher for the other two vehicles. Concentrations of NOx were, for all three vehicles, higher on the road (238-382 ppm) than in the laboratory (129-276 ppm). This is in agreement

with positive acceleration data of Table 2: Our on-road trips resulted more severe than the WMTC and less severe than the R47 legislative cycles due to a combination of trip composition and driving behaviour.

The on-road emission factors calculated with the Mini-PEMS assuming variable volumetric efficiency relative to the laboratory were higher for HC from Vehicle 1 (0.29-0.43 g/km on the road, 0.19-0.20 g/km laboratory) likely due to lower ambient temperature and comparable for the other two vehicles. CO emissions in the laboratory were higher from Vehicle 1

due to prolonged wide open throttle operations and lower for the other two vehicles which instead could easily follow the WMTC speed trace. NOx emissions in the lab and on the road were comparable for Vehicle 1 and 3, but larger on the road for Vehicle 2 (0.11-0.14 g/km laboratory, 0.23-0.35 g/km on the road). The emissions of $CO_2$ on the road were lower (48-51 g/km Vehicle 1, 39-45 g/km Vehicle 3) or comparable (72-79 g/km Vehicle 2) to the laboratory tests (55-66 g/km Vehicles 1 and 3, 75-82 g/km Vehicle 2. The particle mass emissions on the road (0.35-0.44 g/km) were comparable to slightly higher

than in the laboratory (0.18-0.40 g/km as measured by Mini-PEMS), while particle number emissions (including volatiles) on the road (3.4-4.4 x $10^{11}$ particles/km) were comparable to slightly lower relative to the 5.5 x $10^{11}$ – 1.5 x $10^{12}$ particles/km measured by the Mini-PEMS in the laboratory (Table S8).

As an example, the instantaneous emission rates of HC, CO and NOx are given in Figure 9 for Vehicle 2. The sharp accelerations after a stop are clearly visible in terms of HC and CO emissions: while HC large emissions decrease after few

seconds of acceleration, the CO emissions remain high during wide-open throttle operations, indicating that this mode of driving (potentially frequent on small motorcycles) dominates the amount of CO emissions.

A comparison of emissions between laboratory and on-road trips both calculated with the Mini-PEMS is plotted in Figure 10; laboratory results are averaged on cold and hot start tests for readability, detailed results can be found in Table 6. Pollutants real driving emissions are generally larger than during legislative tests with HC and CO deviations enhanced




during on-road cold start tests and larger $NO_x$ deviations during hot start tests. One exception is the large CO emission factor of Vehicle 1 in the laboratory for the reasons explained above. The behaviour of $CO_2$ emissions, in the opposite direction with larger values related to laboratory tests for Vehicle 1 and 3, indicated better after-treatment operations in the lab than on-road as well as smaller fuel consumption on-road for the 2 vehicles. This is supported by the comparison between fuel

5   consumption on-road and on the roller bench for Vehicle 2 (Table 6 and Table 7, driving distance of WMTC = 13.1 km): a comparable fuel consumption is related to comparable $CO_2$ emissions (Figure 10).

In conclusion, the Mini-PEMS can be deployed on-road with minimum effort and little safety precautions (secured rack and fasten cables) and has enough resolution to distinguish different test severity, driving behaviour, and cold/hot engine start conditions.



# 4 Conclusions

We presented an exploratory study aiming at assessing miniature portable emission measuring system (Mini-PEMS) suitable to be installed on-board of 2-wheelers (such as scooters and motorcycles) given its small size and weight. The Mini-PEMS, designed at the Technical University of Liberec, is capable of measuring the exhaust concentrations of hydrocarbons (HC),

carbon monoxide (CO) and dioxide ($CO_2$), and nitrogen oxides (NO and $NO_2$) using non-dispersive infrared and electrochemical cell techniques. In addition, it acquires a series of engine and vehicle parameters such as exhaust and inlet air temperatures, manifold absolute pressure, engine and vehicle speed and GPS coordinates of on-road trips. These are used by the Mini-PEMS to calculate the exhaust flow and the emission factors in mass/distance units.

The Mini-PEMS was tested on three 2-wheelers (1 moped and 2 motorcycles) and compared with standard bench

instrumentation inside the emission test cell of the European Commission Joint Research Centre (JRC, VELA laboratories) during legislative driving cycles. As an application, the Mini-PEMS was deployed on-board of the same vehicles during repeated on-road trips in order to measure real-driving emissions. The mean absolute deviations of exhaust gas concentrations were 8% for HC (range -17 to +16%), 8% for CO (-3 to +19%), 13% for $NO_x$ (-12 to- +24%) and 2% for $CO_2$ (-7 to +5%) In terms of emission factors, which include the exhaust flow uncertainty, the differences were 7% (-17 to +10%)

for HC, 7% (-4 to +14%) for CO, 9% (-22 to +2%) for NOx and 5% (-4 to +11%) for $CO_2$. For one vehicle where fuel consumption was measured directly with a dedicated fuel flow meter, the absolute deviations between fuel consumption over the legislative driving cycle measured in the laboratory and calculated by the Mini-PEMS with a carbon balance method (mass emissions of HC, CO and $CO_2$) were in the range 1-3%.

On-road tests were not designed to be representative of typical operation of 2-wheelers (which at present are neither agreed

nor proposed in the scientific literature and EU legislation). The real-driving emissions had larger variability than in the laboratory (up to a factor of 2 against typically 10%). HC and NOx emissions on-road were similar or larger than in the laboratory, CO emissions were larger on-road for the motorcycles and smaller for the moped (because of shorter periods of high engine load in the laboratory). At present, it is not possible to draw general conclusions on the real-driving emission results as they were vehicle- and compound- specific, but there are indications on what to focus on for future studies, namely

HC emissions after cold engine start or cold ambient conditions and CO emissions during prolonged periods of high engine loads.

All in all, the results are satisfactory considering that (i) the size and weight of the Mini-PEMS is smaller than standard PEMS instruments deployed in passenger cars, (ii) the detectors are different, simpler and cheaper, (iii) the external gas line is unheated, (iv) the exhaust flow is calculated and not directly measured, (v) the exhaust flow is not kept at 190 ℃ inside the

instrument, (vi) large vibrations and the presence of frequent bumps on the road. The Mini-PEMS has certainly the potential to be employed for research and development purposes, in periodical road worthiness tests when excessive emissions indicate malfunctioning of after-treatment devices and in the preliminary phases of an in-service monitoring.



# Author contribution

MV, LD and MP have designed and fabricated the Mini-PEMS.

MV performed the Mini-PEMS data analysis and prepared the manuscript.

AAZ performed the analysis of roller bench emissions data, drove the vehicles outdoor, and contributed to text and Figures.

LD and MP adapted the vehicles and installed the Mini-PEMS on-board.

FF, FM, MC assisted during Mini-PEMS installation and followed the vehicle on-road for safety reasons.

BG and MG supervised the experimental activity, reviewed and edited the manuscript.

# Disclaimer for the authors affiliated with the European Commission

The opinions expressed in this manuscript are those of the authors and should in no way be considered to represent an

official opinion of the European Commission. Mention of trade names or commercial products does not constitute

endorsement or recommendation by the authors.

# Acknowledgements

The authors would like to thank the technical staff of VELA and in particular Gaston Lanappe and Philippe Le Lijeour .

Visits of the authors from the Czech Republic were funded in part by the Technical University of Liberec staff mobility grant

and by the Czech Ministry of Education projects LO1311.





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



# Tables

**Table 1. Vehicles' technical specifications. All vehicles were equipped with a 4-stroke spark-ignition engine and a constant variable transmission. The reported driving cycle is the actual test cycle used in this work, and not the type-approval; see text for details. Notes: [1]Type and Category according to Regulation EU No.134 (EC, 2014); [2]MP = medium performance; [3]2w = 2-way oxidation catalyst, 3w = 3-way catalyst; [4]carb = carburettor, inj = injection; [5]WOT = wide-open throttle.**

| Parameter | | Vehicle | | |
|---|---|---|---|---|
| | | 1 | 2 | 3 |
| Type[1] | | Moped | MP[2] Motorcycle | MP Motorcycle |
| Category | | L1e-B | L3e-A2 | L3e-A2 |
| Capacity | [cm$^3$] | 50 | 300 | 150 |
| Power | [kW] | 2.5 | 16 | 9.5 |
| Mileage | [km] | 4500 | 1500 | 1200 |
| Year | | 2010 | 2015 | 2015 |
| Emissions | | Euro 2 | Euro 3 | Euro 3 |
| Mass | [kg] | 85 | 180 | 130 |
| After-treatment[3] | | 2w | 3w | 3w |
| Fuel system[4] | | carb | inj | inj |
| Cycle[5] | | R47/WOT | WMTC 2-2 | WMTC 2-1 |



**Table 2. Roller bench driving cycles and on-road trip parameters. Notes:** [1]**Calculated from speed profiles as in UNECE Regulation No. 47 and Regulation EU 134/2014 (EC 2014).**

| Parameter | Driving cycles [1] | | | On-road trip | | |
|---|---|---|---|---|---|---|
| | ECE R47 | WMTC 2-1 | WMTC 2-2 | Vehicle 1 | Vehicle 2 | Vehicle 3 |
| Total distance [m] | 6 259 | 12 287 | 13 177 | 2 242 | 2 245 | 2 240 |
| Total time [s] | 896 | 1 200 | 1 200 | 398 | 189 | 235 |
| Drive time [s] | 776 | 1 041 | 1 046 | 368 | 186 | 218 |
| Temperature [şC] | 25 | 25 | 25 | 7 | 15 | 22 |
| Average driving speed [km/h] | 25.1 | 36.9 | 39.5 | 21.1 | 42.8 | 35.0 |
| Maximum speed [km/h] | 45.0 | 82.5 | 94.9 | 40.5 | 82.5 | 71.90 |
| Speed [km/h] [$25^{th}$, $75^{th}$] percentile | [20.0,42.8] | [21.9,55] | [23.5,59.5] | [13.8,30.8] | [25.2,60.7] | [20.5,52.8] |
| Average positive acceleration [m/s$^2$] | 1.25 | 0.42 | 0.47 | 0.32 | 1.02 | 1.02 |
| Positive acceleration [m/s$^2$] [$25^{th}$, $75^{th}$] percentile | [1.26,1.26] | [0.11,0.66] | [0.11,0.61] | [0.12,0.34] | [0.56,1.42] | [0.19,1.66] |



**Table 3. Mini-PEMS technical details.**

| Compound/ Parameter | Method | Range/Value | LOD ([1]) | $T_{0-90}$ [s] |
|---|---|---|---|---|
| HC | NDIR | 0-24000 ppmC | 14 ppmC | 2-3 |
| CO | NDIR | 0 - 12% | 0.004% | 2-3 |
| $CO_2$ | NDIR | 0 - 20% | < 0.01% | 2-3 |
| NO | Electrochemical cell | 0 - 5 000 ppm | 3 ppm | 3-5 |
| $NO_2$ | Electrochemical cell | 0 - 300 ppm | 1 ppm | 15-30 |
| PN([2]) | Ionization chamber | $\approx 10^7$ #/cm$^3$ | $\approx 5 \times 10^5$ # /cm$^3$ | 5-10 |
| MAP | Pressure Transducer | 0 - 250 kPa abs. | N/A | < 1 |
| rpm | Coil pickup | 0 - 20 000 min$^{-1}$ | N/A | < 1s |
| | Vibration sensor | | | |
| | Optical sensor | | | |
| Size No. 1 | | 45 x 31 x 18 cm | | |
| Weight No. 1 | | 13 + 4.7 + 3.4 kg | | |
| Size No. 2 | | 40 x 20 x 20 cm | | |
| Weight No. 2 | | $\approx$ 10 kg | | |

([1])     LOD – limit of detection, calculated as 3 x standard deviation of noise when sampling ambient air

([2])     Particle Number from total particle length.





**Table 4. Comparison of HC, CO, NOx and CO$_2$ raw exhaust concentrations ([ppm]) measured with the standard test cell instrumentation (bench) and the Mini-PEMS No.1 during legislative test cycles (WMTC and R47) and wide open throttle tests (Vehicle 1) on the roller bench. Dev = deviation; MAPD = mean absolute percentage deviation; min/max = minimum/maximum deviations. ([1]) Tests performed with additional Mini-PEMS No. 2.**

| Test | Veh. | Cycle | Start | HC Bench [ppm] | HC PEMS [ppm] | HC Dev [%] | CO Bench [ppm] | CO PEMS [ppm] | CO Dev [%] | NO Bench [ppm] | NO PEMS [ppm] | NO Dev [%] | CO$_2$ Bench [ppm] | CO$_2$ PEMS [ppm] | CO$_2$ Dev [%] |
|---|---|---|---|---|---|---|---|---|---|---|---|---|---|---|---|
| 1 | 1 | R47 | Cold | 1744 | 1455 | -17 | 8289 | 8417 | 2 | 222 | 276 | 24 | 119111 | 124998 | 5 |
| 2 | 1 | R47 | Hot | 1673 | 1550 | -7 | 11657 | 13263 | 14 | 177 | 171 | -3 | 118131 | 119537 | 1 |
| 3 | 1 | WOT | Hot | 3215 | 3722 | 16 | 14051 | 14549 | 4 | 252 | 220 | -12 | 115970 | 108021 | -7 |
| 4 | 1 | WOT | Hot | 1978 | 2089 | 6 | 13189 | 12741 | -3 | 217 | 194 | -11 | 113697 | 113629 | 0 |
| MAPD | 1 | R47,WOT | - | - | - | 11 | - | - | 6 | - | - | 13 | - | - | 3 |
| 5 | 2 | WMTC | Cold | 789 | 778 | -1 | 2201 | 2460 | 12 | 151 | 181 | 20 | 137897 | 136604 | -1 |
| 6 | 2 | WMTC | Cold | 837 | 803 | -4 | 1945 | 2098 | 8 | 134 | 146 | 9 | 141598 | 140958 | 0 |
| 7 | 2 | WMTC | Hot | 309 | 286 | -8 | 1277 | 1346 | 5 | 119 | 129 | 9 | 141007 | 140177 | -1 |
| MAPD | 2 | WMTC | - | - | - | 4 | - | - | 8 | - | - | 12 | - | - | 1 |
| 8 | 3 | WMTC | Cold | 1134 | 1197 | 6 | 3626 | 4325 | 19 | 160 | 189 | 18 | 141634 | 141173 | 0 |
| MAPD | All | All | - | - | - | **8** | - | - | **8** | - | - | **13** | - | - | **2** |
| min | All | | - | - | - | -17 | - | - | -3 | - | - | -12 | - | - | -7 |
| max | All | | - | - | - | 16 | - | - | 19 | - | - | 24 | - | - | 5 |
| 5([1]) | 2 | WMTC | Cold | 789 | 757 | -4 | 2201 | 2329 | 6 | 151 | 172 | 14 | 137897 | 136858 | -1 |
| 6([1]) | 2 | WMTC | Cold | 837 | 568 | -32 | 1945 | 2140 | 10 | 134 | 156 | 16 | 141598 | 142296 | 0 |
| 7([1]) | 2 | WMTC | Hot | 309 | 238 | -23 | 1277 | 1345 | 5 | 119 | 128 | 8 | 141007 | 140796 | 0 |
| MAPD | 2 | WMTC | - | - | - | **20** | - | - | **7** | - | - | **13** | - | - | **0** |





**Table 5. Fuel consumption calculated from laboratory instrumentation (bench) and Mini-PEMS compared to the KMA fuel flowmeter (reference). Volumetric and mass consumption from the bench were derived from Regulation EU 134/2014 (carbon balance, diluted emissions) and from instantaneous mass emissions, respectively.**

| Test | KMA | | Bench | | | | PEMS-1 | | PEMS-2 | |
|------|-----|-----|-------|------|-----|------|--------|------|--------|------|
| | [mL/test] | [g/test] | [mL/test] | Dev% | [g/test] | Dev% | [g/test] | Dev% | [g/test] | Dev% |
| Test 5 | 446 | 330 | 415 | -7 | 303 | -8 | 320.5 | -3 | 324.2 | -2 |
| Test 6 | 460 | 341 | 422 | -8 | 308 | -10 | 339 | -1 | 345.8 | 1 |
| Test 7 | 437 | 324 | 408 | -7 | 288 | -11 | 316.2 | -2 | 316.8 | -2 |



**Table 6. Comparison of emissions factors measured with the standard test cell instrumentation (bench) and the two Mini-PEMS during legislative test cycles (WMTC and R47) on the roller bench. Variable volumetric efficiency was assumed (see text for details). (1)Tests performed with the additional Mini-PEMS No. 2.**

| Test | Vehicle | Cycle | Start | HC | | | CO | | | NOx | | | $CO_2$ | | |
|------|---------|-------|-------|------|------|-----|------|------|-----|------|------|-----|------|------|-----|
| | | | | Bench [g/km] | PEMS [g/km] | Dev [%] | Bench [g/km] | PEMS [g/km] | Dev [%] | Bench [g/km] | PEMS [g/km] | Dev [%] | Bench [g/km] | PEMS [g/km] | Dev [%] |
| 1 | 1 | R47 | Cold | 0.21 | 0.19 | -8 | 2.68 | 2.64 | -1 | 0.14 | 0.14 | 2 | 65.59 | 65.92 | 0 |
| 2 | 1 | R47 | Hot | 0.19 | 0.20 | 4 | 3.93 | 4.02 | 2 | 0.09 | 0.07 | -22 | 59.23 | 56.76 | -4 |
| MAPD | 1 | R47 | - | - | - | **6** | - | - | **2** | - | - | **12** | - | - | **2** |
| 5 | 2 | WMTC | Cold | 0.12 | 0.11 | -2 | 0.81 | 0.93 | 14 | 0.14 | 0.14 | 1 | 72.47 | 77.16 | 6 |
| 6 | 2 | WMTC | Cold | 0.11 | 0.12 | 10 | 0.74 | 0.84 | 13 | 0.13 | 0.12 | -3 | 73.77 | 81.67 | 11 |
| 7 | 2 | WMTC | Hot | 0.06 | 0.05 | -17 | 0.56 | 0.61 | 9 | 0.12 | 0.11 | -7 | 71.86 | 75.15 | 5 |
| MAPD | 2 | WMTC | - | - | - | **9** | - | - | **12** | - | - | **4** | - | - | **7** |
| 8 | 3 | WMTC | Cold | 0.12 | 0.13 | 3 | 0.82 | 0.87 | -4 | 0.13 | 0.11 | -16 | 57.16 | 54.97 | -4 |
| MAPD | All | All | - | - | - | **7** | - | | **7** | - | | **9** | - | - | **5** |
| Min | All | All | - | - | - | -17 | - | - | -4 | - | - | -22 | - | - | -4 |
| Max | All | All | - | - | - | 10 | - | - | 14 | - | - | 2 | - | - | 11 |
| 5([1]) | 2 | WMTC | Cold | 0.12 | 0.11 | -2 | 0.81 | 0.91 | 12 | 0.14 | 0.14 | -3 | 72.47 | 78.08 | 8 |
| 6([1]) | 2 | WMTC | Cold | 0.11 | 0.08 | -24 | 0.74 | 0.91 | 22 | 0.13 | 0.13 | 4 | 73.77 | 83.32 | 13 |
| 7([1]) | 2 | WMTC | Hot | 0.06 | 0.04 | -32 | 0.56 | 0.62 | 10 | 0.12 | 0.10 | -11 | 71.86 | 75.34 | 5 |
| MAPD | 2 | WMTC | - | - | - | **19** | - | - | **15** | - | - | **7** | - | - | **9** |



**Table 7.** Exhaust concentrations (average), real-driving emissions factors and fuel consumption during on-road trips.

| Trip | Vehicle | Start | HC [ppm] | CO [ppm] | NOx [ppm] | $CO_2$ [ppm] | HC [g/km] | CO [g/km] | NOx [g/km] | $CO_2$ [g/km] | Fuel [g/km] |
|---|---|---|---|---|---|---|---|---|---|---|---|
| 1 | 1 | Warm | 3732 | 8895 | 335 | 106491 | 0.378 | 1.885 | 0.159 | 48 | 16.6 |
| 2 | 1 | Hot | 4037 | 5074 | 382 | 113468 | 0.426 | 1.256 | 0.182 | 51 | 17.3 |
| 3 | 1 | Hot | 3603 | 9349 | 328 | 121711 | 0.286 | 1.530 | 0.138 | 48 | 16.3 |
| 1 | 2 | Cold | 803 | 3947 | 276 | 119305 | 0.152 | 2.447 | 0.230 | 79 | 26.5 |
| 2 | 2 | Hot | 554 | 2476 | 271 | 115105 | 0.096 | 1.230 | 0.340 | 75 | 24.5 |
| 3 | 2 | Hot | 521 | 4785 | 371 | 115589 | 0.093 | 2.538 | 0.350 | 72 | 24.3 |
| 1 | 3 | Cold | 2099 | 17088 | 239 | 103994 | 0.256 | 6.218 | 0.104 | 45 | 17.7 |
| 2 | 3 | Hot | 1122 | 13423 | 293 | 101117 | 0.163 | 4.581 | 0.127 | 41 | 15.5 |
| 3 | 3 | Hot | 1078 | 14923 | 238 | 100000 | 0.160 | 5.054 | 0.098 | 39 | 15.1 |





# Figures

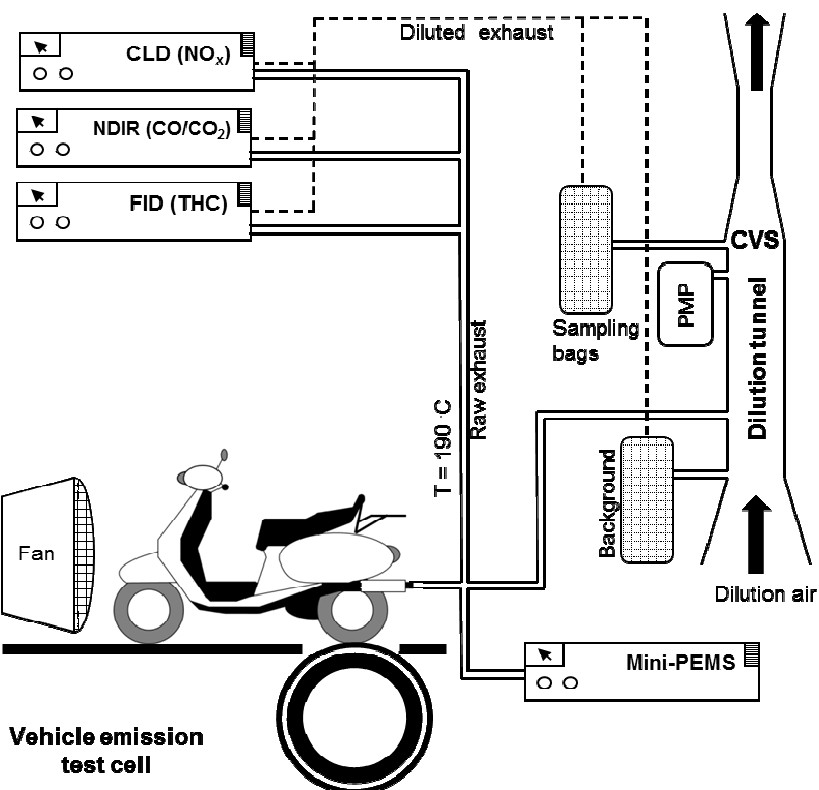

**Figure 1. Schematic of the test cell and laboratory instrumentation. Exhaust emission measurements are performed at the tailpipe and after dilution through a constant volume sampler (flow rate ≈ 5m³/min) by two independent gas analyzer systems. The Mini-PEMS collects part of the raw exhaust; see text for details.**





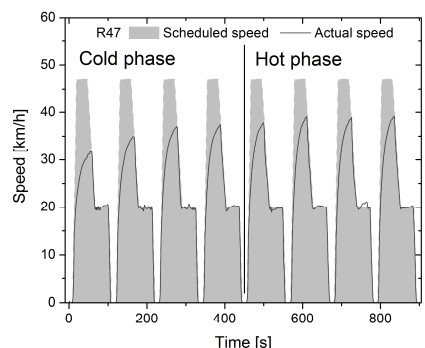
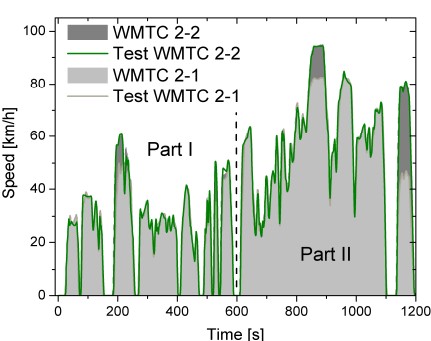

**Figure 2. Legislative speed profiles (grey shaded area) and examples of actual test speed for driving cycles ECE-R47 (left panel), WMTC type 2-1 and WMTC 2-2 (right panel) run by vehicles 1, 3 and 2, respectively; see text for details.**





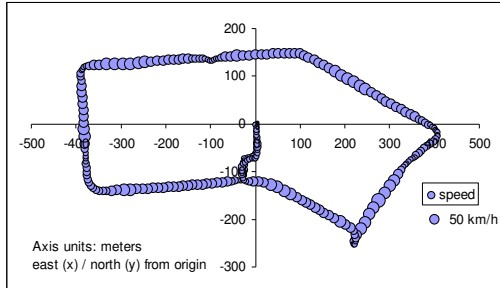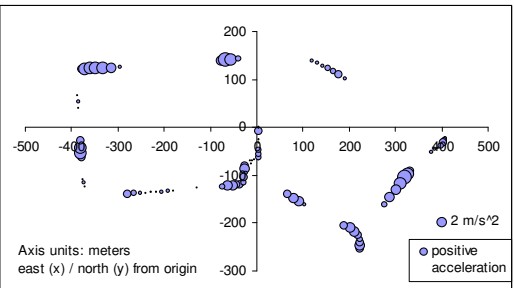

**Figure 3. Top view of on-road trips geometry ([m]) with size-coded speed (left panel) and positive acceleration (right panel) for Vehicle 2. The vehicles had to pass 3 stop road-signals and 2 roundabouts.**



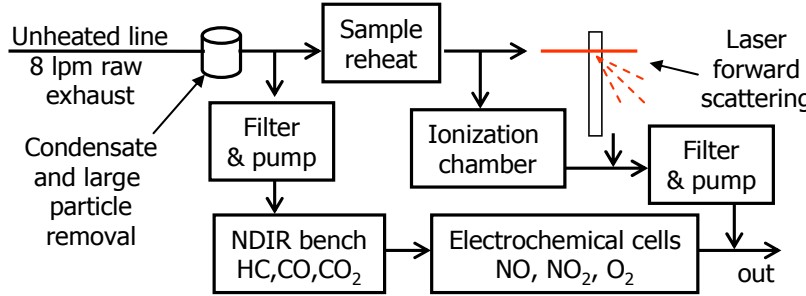

**Figure 4. Schematic of the miniature portable on-board emissions monitoring system (Mini-PEMS); see text for details.**





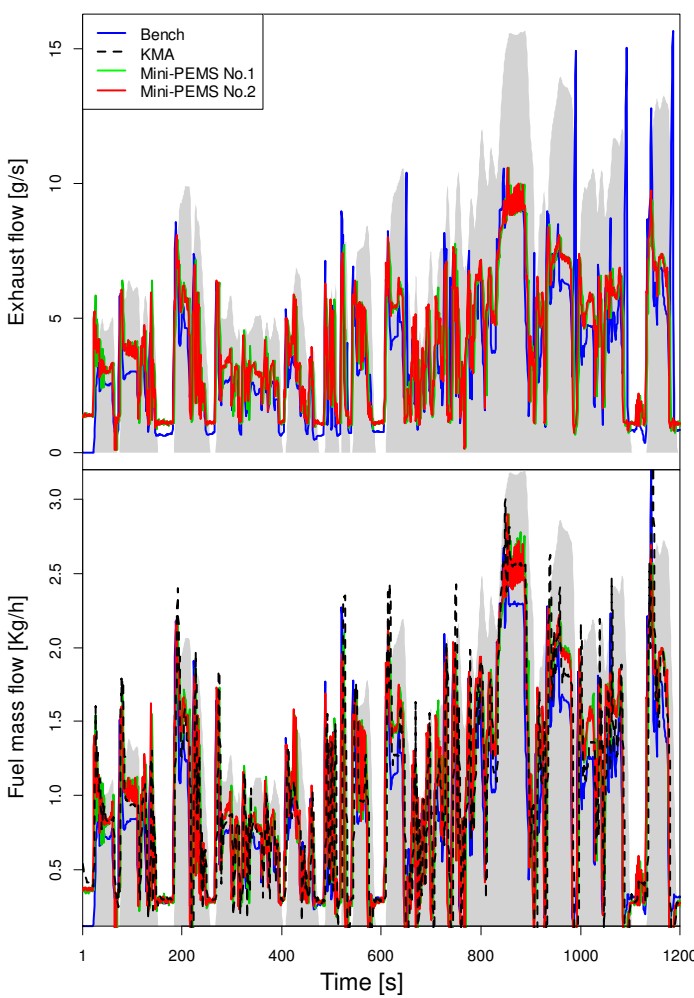

**Figure 5. Upper panel: Comparison of the exhaust flow from bench instrumentation calculated with the $CO_2$ tracer method and from the 2 Mini-PEMS during Test 5. Lower panel: Comparison of fuel consumption (i) measured by a dedicated fuel flowmeter (KMA); (ii) calculated with the carbon balance method using HC, CO and $CO_2$ mass emissions rates from bench instrumentation; and (iii) calculated from Mini-PEMS No. 1 and No. 2.**

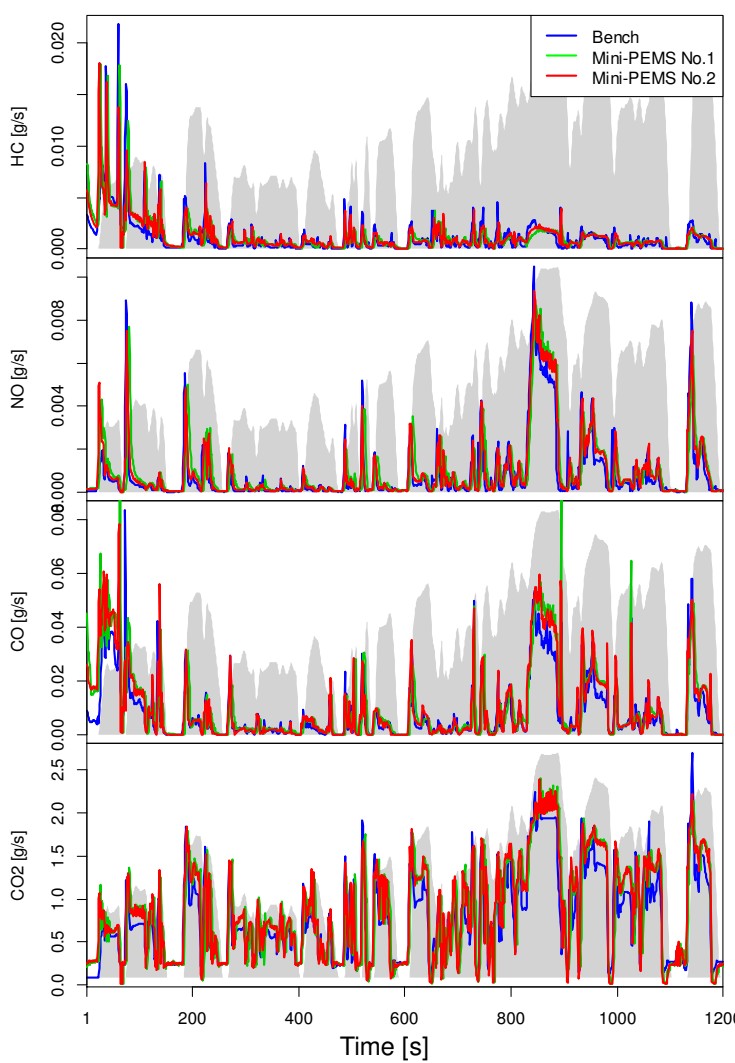

**Figure 6. Mass flow rates of HC, NOx, CO and CO$_2$ from bench instrumentation and from the two Mini-PEMS for Vehicle 2 during Test 5 (WMTC).**


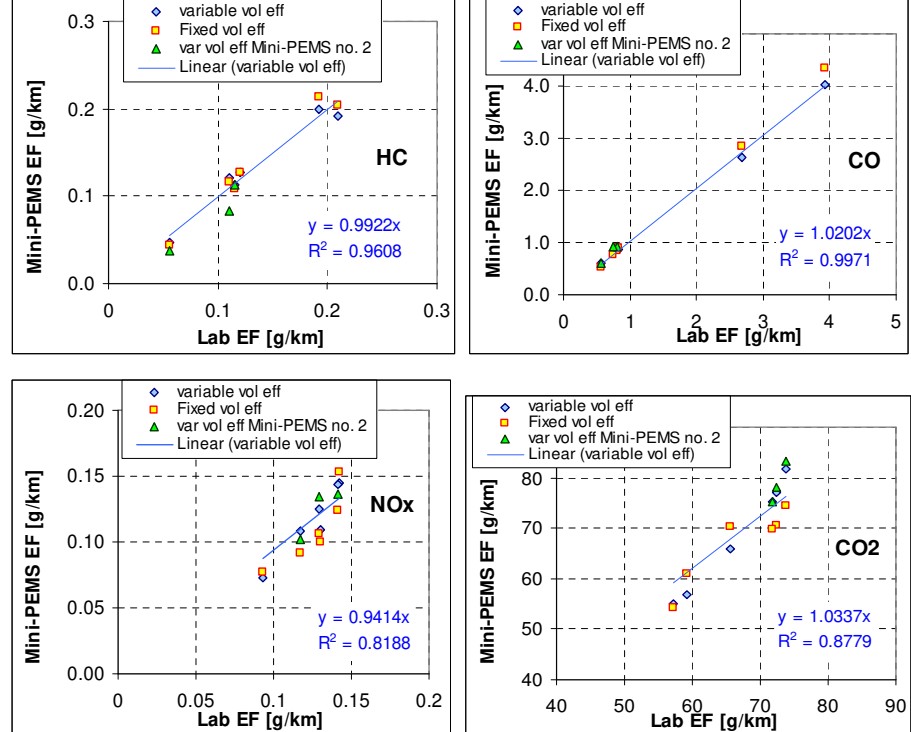

**Figure 7. Emission factors of HC, NOx, CO and CO₂ from bench instrumentation (horizontal axis), and from Mini-PEMS No. 1 (vertical axis) obtained during roller bench tests.**



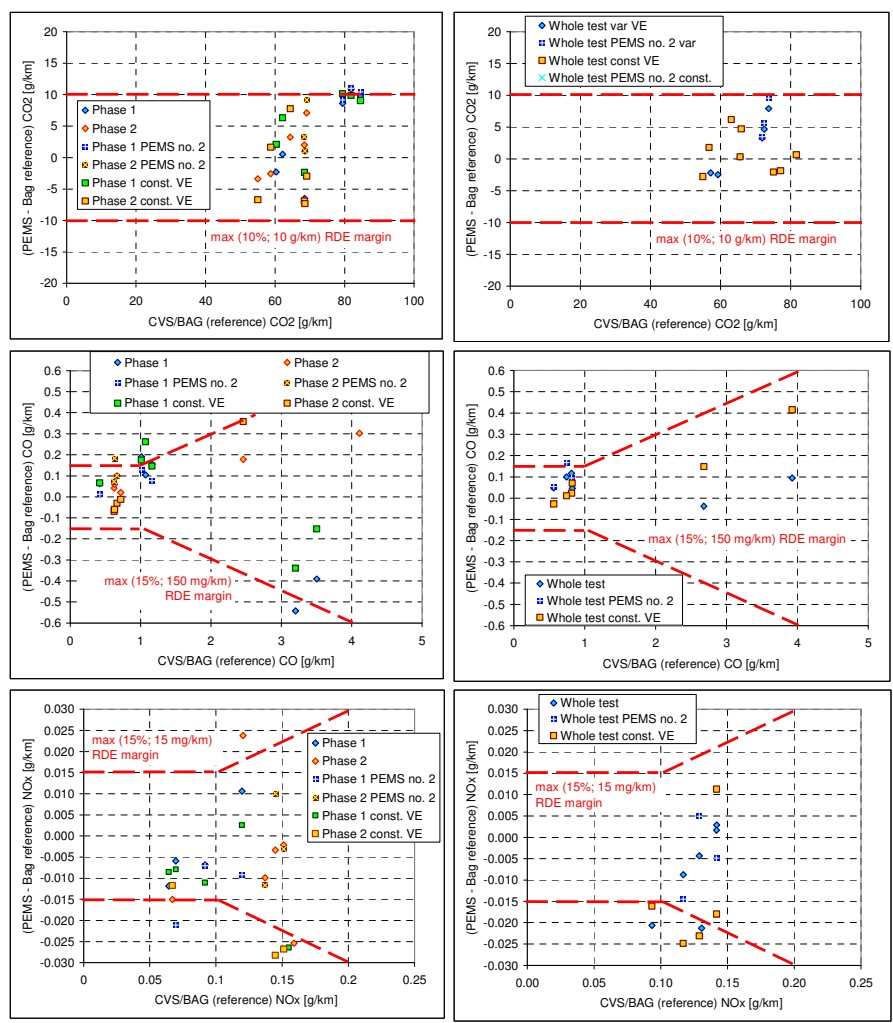

**Figure 8. Relative deviations (%) between PEMS and laboratory emission factors during roller bench tests. Red dashed lines represent the PEMS validation tolerances as per Regulation EU 2018/1832 (EC, 2018a); see text for details.**





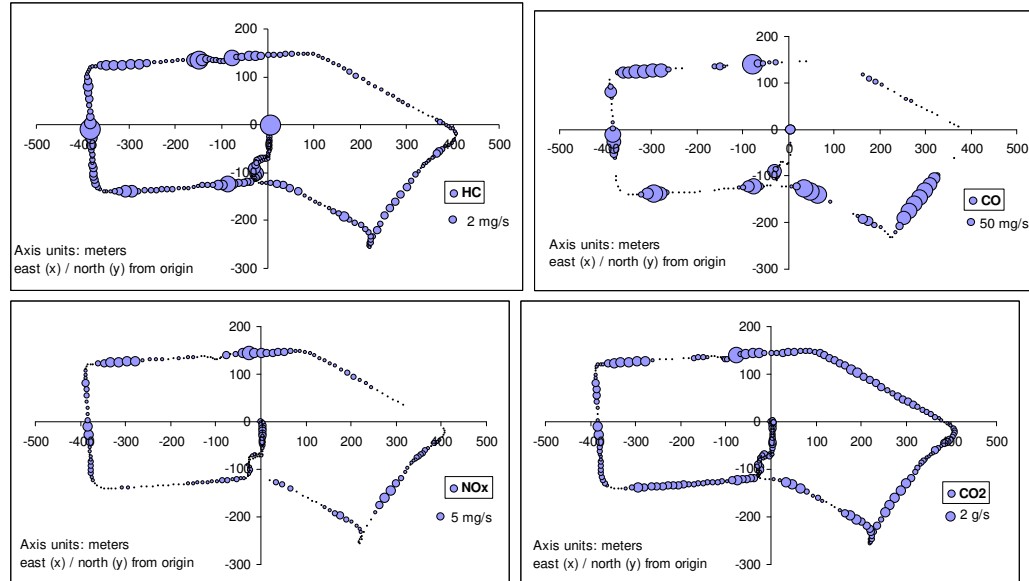

**Figure 9. Top view of the on-road trip geometry with size-coded instantaneous mass emissions rates of HC, CO, NOx and CO₂ averaged on three runs of Vehicle 2. See details in Figure 3).**





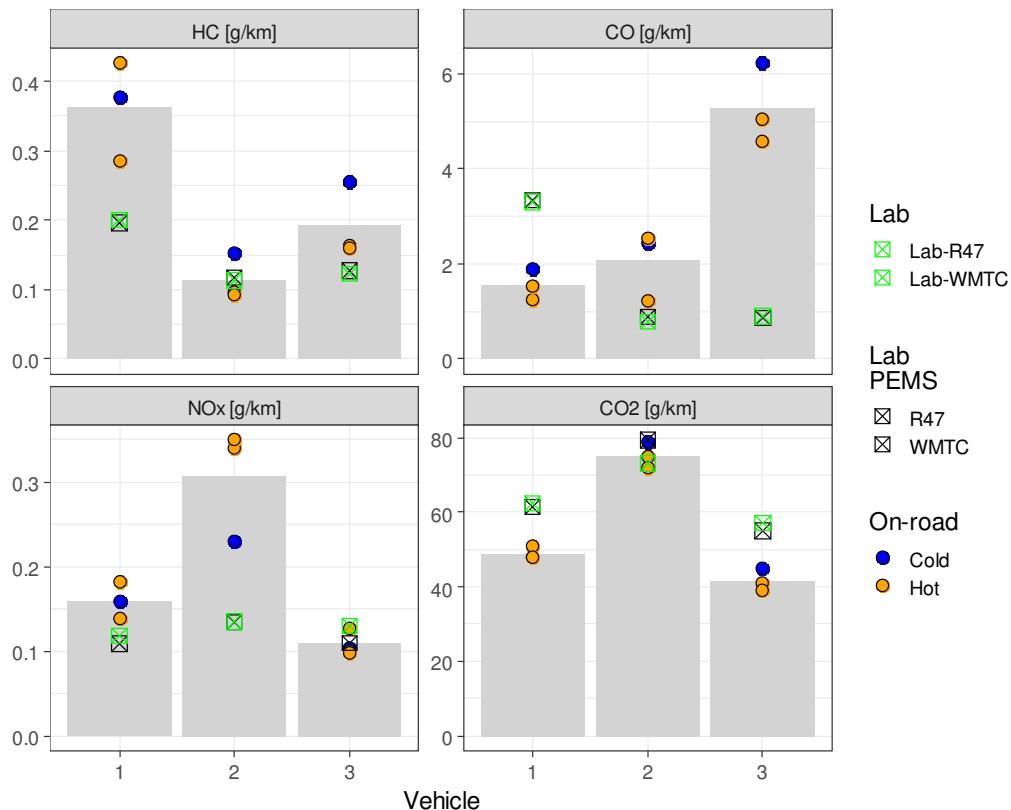

Figure 10. Summary of real driving and laboratory emissions. Roller bench data points were obtained with laboratory instrumentation (green) and the Mini-PEMS (black). Grey bars are average real driving emission factors obtained from single on-road cold start (blue) and hot start tests (orange).