# Peer review of "A miniature Portable Emissions Measurement System (PEMS) for real-driving monitoring of motorcycles"

_Atmospheric Measurement Techniques, 2019_

## Referee Comment (RC1) · Anonymous Referee #3 · 11 May 2020

The manuscript is fit for AMT but is quite long and at times reads a little bit like a report rather than an easy to read manuscript. Some parts discuss observations rather than having a focus on measurement techniques, the interesting part of lab vs on road PEMS for motorcycles gets a little bit too much in the background and could stand out more.

I have mainly minor comments below and suggest may be refocusing the manuscript by putting some of the technical tables that convey only information but no results in the SI (you are starting results with table 4). I would also put some of the figures in SI such as the road trips (Figs 3 and 9). This will help shorten the manuscript and make

it more readable.

Details

Please tighten experimental section as some things are repetitive (e.g. twice the size cut of your TSI CPC).

Formulas page 7 and 8 appear in a poor resolution.

Figure 5: data in bright green is hardly visible except in a few plac.es

Figure 7: This figure should be substantially improved. Please homogenize the panels. Do not use frames for the panels. Please align the axis between panels, make sure the legend and the axis units do not overlay etc. . . Also please use subscripts on chemical formulas. Please add error bars to his figure, indicating measurement uncertainties, or at least indicate them on one point.

Figure 8: same as figure 7 with attention to detail and quality. Please put legends in similar spots, align axis, get rid of frames, add error bars,. . .

Figure 10: please add units to axis rather than in the title. Also this bright neon green is not well visible. Again error bars should be provided, this is an analytical journal.

---

## Referee Comment (RC2) · Anonymous Referee #4 · 29 May 2020

General comments: This paper talks about tests to evaluate a new miniaturized portable measurement system (Mini PEMS) specifically design to be applied in motorcycles. It is shown the principles of measurement, comparison tests from laboratory and some results from road tests. The article is a little bit long but very interesting since it is bringing to light a usually not-so-discussed issue that is the pollution coming from motorcycles, that if in Europe can be not so significant, in another hand in Asia and Latin America the powered-2-wheel fleet is relevant, as well their emission. It is interesting also the concept of a PEMS that can be assembled in a motorcycle. Although a commercial PEMS can do it, the high weight and high cost make the practical utilization impossible. The scientific approach done in the paper is correct but some points

can be improved: - The general objective is not clear. Is it maybe to validate the Mini PEMS? Or is it just to demonstrate the new technology? - It was a little bit confusing when in the paper it was introduced one Mini PEMS concept; after it was shown a second one, described as "the same technology" but clearly it was a different instrument. Some evaluations are focused on the Mini PEMS #1, others in #2 and some in both.

Specific comments: - Page 4, line 21 to page 5, line 2: this description could be summarized by just saying which method was applied. - Page 6, line 5 to 9: why "real-driving test" doesn't follow minimally the RDE procedure? It could be done just the urban trip; it would be much more interesting and would be useful to compare with car emissions also. - Page 6, lines 20 to 28: It is not clear what particulate size is being measured here. Is there a way to separate them to measure PM10 or PM2.5? Another point is that the laboratory method measures just non-volatile particles and Mini PEMS is measuring everything, thus the results are not comparable. - Page 7, lines 5 to 8: Pictures referenced here are confused, with many details to see in a small picture and first it is shown the details (Picture S1) and the general assembly after (Picture S2). I suggest splitting these pictures in order to be bigger and placed in sequence. - Page 7, line 26: it is not true that motorcycles engines have smaller rotational speeds than a car, usually they run about two times faster than a four-cylinder car engine. - Page 7, line 27: Equation 2: where is considered the engine displacement? - Page 12, lines 15 to 28 and supplementary Table S5: I'm not sure that it is possible to consider the Mini PEMS has a good agreement to laboratory bench because variations showed here are larger than allowed to a regulatory PEMS. - Page 12, line 20: THC is not a regulated pollutant but all regulations regarding RDE in EU have instructions about THC measurements, so it is important to keep it in sight, even to demonstrate a weakness in the Mini PEMS - Page 12, line 20, Figure 8: this figure shows only results from Mini PEMS #2. What about Mini PEMS #1? Is this paper evaluating the one, two, or both? - Supplementary, Figure S4: It is not clear what is represented in these two graphics. Is it for tests 1 and 2? I suggest identifying them, even because in Table S6 is described three tests.

Technical corrections: - I don't understand why tables and figures are at the end of the paper. If it is an editor requirement, ok; if not, it will be easier to follow the authors' thought if figures and tables are inserted into the text. - Equations 1, 2, 3 and 4: poor resolution; it is missed a space (white line) before and after equation 4 - Page 12, line 31: Text makes reference to Table S8 but I believe that the correct is Table S6 (also for page 14, line 28) - Page 27, Table 6: It would be interesting to show three significant algharisms instead of only two for all pollutants except CO2, e.g., HC on the bench, test 1, 0.209 instead of 0.21.

---

## Author Comment (AC1) · 2 Jul 2020

The manuscript is fit for AMT but is quite long and at times reads a little bit like a report rather than an easy to read manuscript. Some parts discuss observations rather than having a focus on measurement techniques, the interesting part of lab vs on road PEMS for motorcycles gets a little bit too much in the background and could stand outmore.

** We thank the reviewer for the comments. We tried to address all of them (replies are marked with "**" as the applilcation does not allow text formatting. **

I have mainly minor comments below and suggest may be refocusing the manuscript by putting some of the technical tables that convey only information but no results in the SI (you are starting results with table 4). I would also put some of the figures in SI such as the road trips (Figs 3 and 9). This will help shorten the manuscript and make it more readable.

Details Please tighten experimental section as some things are repetitive (e.g. twice the size cut of your TSI CPC).

** The experimental section has been shortened by moving Fig. 2. Fig. 3 and Table 2 to the SI file, and by shortening the text describing particulate matter measurements (page 5 lines from 3 to 13 of the original manuscript). The remaining part of the experimental section is, in our opinion, relevant to the results presented. **

Formulas page 7 and 8 appear in a poor resolution.

** All equations in the manuscript have been rewritten. **

Figure 5: data in bright green is hardly visible except in a few places

** We have improved Figure 5 for better readability. **

Figure 7: This figure should be substantially improved. Please homogenize the panels.Do not use frames for the panels. Please align the axis between panels, make sure thelegend and the axis units do not overlay etc...Also please use subscripts on chemical formulas. Please add error bars to his figure, indicating measurement uncertainties, or at least indicate them on one point.

** The Figure has been improved as suggested. **

** The estimate of the measurement uncertainty information has been added to the text at page 11 line 22 of the new version: "The estimated measurement uncertainty for the laboratory measurement (bag data) of NOx is 10% at 80 mg/km and 5% at the 150 mg/km emission level, of HC is 10% at 50 mg/km and 5% at 200 mg/km, and

of CO and CO2 is 3% for all emission levels. This uncertainty is shown in Figure 6 as a separate arrow rather than being added to each point. The largest components of uncertainty for the Mini-PEMS are the engine volumetric efficiency (affecting the exhaust flow calculations), the uncertainty of HC and PM measurements due to the limitations of the approach chosen (unheated sampling train, surrogate measurements for PM), and the uncertainty associated with dynamic events (rapid changes in both exhaust flow and pollutant concentrations). These non-controllable uncertainties were estimated ex post to be overall in the range from 10% up to 20% (see MAPD, mean absolute percentage differences, in Table 4 and Table 6). The known uncertainty of gaseous component measurements at steady-state, above the detection limit, is 3-5%, and the combined uncertainty of engine rpm and intake manifold pressure and temperature is 1-2%. We estimated this known uncertainty to be about 5% for all measurements." **

Figure 8: same as figure 7 with attention to detail and quality. Please put legends in similar spots, align axis, get rid of frames, add error bars,...

** The Figure has been revised. The estimated uncertainty of laboratory measurement was added as a separate arrow to reflect its overall estimate, rather than calculation specific to each point. **

Figure 10: please add units to axis rather than in the title. Also this bright neon green is not well visible. Again error bars should be provided, this is an analytical journal

** The Figure was improved as suggested. **

---

## Author Comment (AC2) · 2 Jul 2020

\*\* The responses are marked with two asterisks at the beginning and at the end of the response. \*\*

This paper talks about tests to evaluate a new miniaturized portable measurement system (Mini PEMS) specifically design to be applied in motorcycles. It is shown the principles of measurement, comparison tests from laboratory and some results from road tests. The article is a little bit long but very interesting since it is bringing to light a usually not-so-discussed issue that is the pollution coming from motorcycles, that if in Europe can be not so significant, in another hand in Asia and Latin America

the powered-2-wheel fleet is relevant, as well their emission. It is interesting also the concept of a PEMS that can be assembled in a motorcycle. Although a commercial PEMS can do it, the high weight and high cost make the practical utilization impossible. The scientific approach done in the paper is correct but some points can be improved:

** We thank the reviewer for the comments. We tried to address all of them. **

The general objective is not clear. Is it maybe to validate the MiniPEMS? Or is it just to demonstrate the new technology? It was a little bit confusing when in the paper it was introduced one MiniPEMS concept; after it was shown a second one, described as "the same technology" but clearly it was a different instrument. Some evaluations are focused on the Mini PEMS #1, others in #2 and some in both.

** The last paragraph of the introduction has been modified as follows:

"The goal of this study was to demonstrate a miniature PEMS device (Mini-PEMS, here-after) suitable to be fitted on motorcycles, scooters and mopeds, compare it against standard laboratory instrumentation during roller bench tests, and to assess its per-formance during on-road tests of three 2-wheelers, 1 moped and 2 motorcycles. The validation was conducted with the intended use of the Mini-PEMS as a tool for research and development, road worthiness (periodical inspections) and screening tool for the market surveillance (e.g. Regulation EU 2017/1151)."

The evaluation is mainly based on Mini-PEMS no. 1; with several elements including Mini-PEMS no. 2. The 2 instruments are only physically different (appearance, size and weight) but the operational principle stated in the description is exactly the same. The text has been clarified (page 7 line 7 of the original version):

"A second Mini-PEMS system (Mini-PEMS No. 2) based on the same measurement techniques explained above, and using the same key components (except for the light scattering sensor which was omitted), but smaller and lighter than Mini-PEMS No. 1, . . . " **

Specific comments: - Page 4, line 21 to page 5, line 2: this description could be summarized by just saying which method was applied.

** The experimental section has been shortened by moving Fig. 2. Fig. 3 and Table 2 to the SI file, and by shortening the text describing particulate matter measurements (page 5 lines from 3 to 13 of the original manuscript). The remaining part of the experimental section is, in our opinion, relevant to the results presented. **

- Page 6, line 5 to 9: why "real-driving test" doesn't follow minimally the RDE procedure? It could be done just the urban trip; it would be much more interesting and would be useful to compare with car emissions also.

** We agree with the Reviewer that additional steps of elaboration on RDE results are feasible; it was in our minds too. However, being involved in the development of the RDE procedures in the European Union and worldwide, we know how misleading or dangerous this action can be without collecting large statistical samples, and not only because it could be taken as the approach suggested by researchers at the European Commission. At present, there is no legislative RDE procedure for motorcycles in EU, not even at a discussion phase. There is RDE for passenger cars and RDE for heavy duty engines (The reason was also that a reliable PEMS could not fit a motorcycle, now we demonstrate it can). For instance, the decision of what is the "urban" part of an RDE trip for a motorcycle would be in this manuscript completely arbitrary and changing the value of the urban speed or speed trip composition (typical parameters in RDE) by few kilometers would give different results with no guarantee that a specific speed is more or less representative for that class of L-category vehicles on the road. There are other examples of this kind which can be extracted from the RDE webpage of the United Nations to which our Institutions contributed (www.unece.org). Finally, there is no "average car" to compare to, and also this choice would be completely arbitrary. This is why we avoided further arbitrary elaboration in this direction. We are engaged in the collection of more data in order to do so. Please consider also the scope of the Journal and of this manuscript. **

- Page 6, lines 20 to 28: It is not clear what particulate size is beingmeasured here. Is there a way to separate them to measure PM10 or PM2.5?

** The general size of the particles in engine exhaust ranges from units to hundreds of nm, with very few larger particles. Typically, only the lower size limit is reported, as the upper size limit is not particularly relevant, and PM1, PM2.5 and PM10 are comparable.

We added the following text in the Results section(page 13 line 10 of the new version): "Light scattering is typically sensitive only to particles with a diameter over 100 nm, with the response being dominated by larger particles; on the ionization chamber, there is no lower limit on the particle size. While there is no upper limit for the particle diameter detected, as few particles are larger than 1 $\mu$m, particles larger than several $\mu$m are likely to be trapped in the condensation bowl." **

Another point is that the laboratory method measures just non-volatile particles and Mini PEMS is measuring everything, thus the results are not comparable.

** The inclusion of non-volatile fraction is acknowledged (page 12, line 30-32 of the original manuscript) and comparison data provided (end of chapter 3.1.4. and associated supplemental info). The legislative method (PMP) measures non-volatile particles. For research purposes, non-volatile particle are often included as they are relevant to human health and aerosol chemistry For completeness, we added PM filter data and total particles that we had from previous tests with these vehicle and compared them with the estimated mass and number from the mini-PEMS in Table S9. **

- Page 7, lines 5 to 8: Pictures referenced here are confused, with many details to see in a small picture and first it is shown the details (Picture S1) and the general assembly after (Picture S2). I suggest splitting these pictures in order to be bigger and placed in sequence.

** Pictures placed in a sequence. S1 split and enlarged. **

- Page7, line 26: it is not true that motorcycles engines have smaller rotational speeds

than a car, usually they run about two times faster than a four-cylinder car engine.

** We agree with the Reviewer, but the manuscript does not say that. The manuscript says that the volumetric efficiency of motorcycle engines is considerably lower than that of automobile engines, with this effect being more pronounced at low engine speeds.

The text has been reworded for added clarity (page 7 line 17 of the new version): "Empirical adjustments were therefore introduced based on laboratory tests to account for considerable lower volumetric efficiencies for small motorcycle engines, especially in the lower end of their rpm range." **

- Page 7,line 27: Equation 2: where is considered the engine displacement?

** Thanks for noting this: It was omitted by mistake and has been added to the equation. **

- Page 12, lines 15 to 28 and supplementary Table S5: I'm not sure that it is possible to consider the MiniPEMS has a good agreement to laboratory bench because variations showed here are larger than allowed to a regulatory PEMS.

** We thank the Reviewer for this comment as it allows explaining better. Variations are at times larger than allowed for a regulatory PEMS, which is designed for cars, not for motorcycles. There is no tolerance defined for motorcycle RDE, as there is no RDE legislation for L-cat vehicles. The MiniPEMS was not designed nor certified as a regulatory PEMS for cars, an instrument which typically costs and weighs about 10 times more and deploys different techniques (hence excluded from application on motorcycles or small vehicles). Also, it is plausible that the variation tolerance for motorcycles PEMS will be larger than for cars, if and when legislation in this sense will be drafted. The reasons are the intrinsic Mini-PEMS larger measurement uncertainty, the large pulsation of the exhaust flow from 1 cylinder engines, and the large distribution of emissions in repeated cold starts [see e.g. Zardini et al. Transport Procedia, 14, 2016 and references therein]. The good results (< 20% discrepancy) or very good results (<

10% discrepancy) against regulatory methods indicate acceptable performance in the scope of the instrument. **

- Page 12, line 20: THC is not a regulated pollutant but all regulations regarding RDE in EU have instructions about THC measurements, so it is important to keep it in sight, even to demonstrate a weakness in the Mini PEMS - Page 12, line 20,

** We agree with the Reviewer. RDE should and very likely will include THC measurements in the future, especially in the case of motorcycles or small mopeds. This is the reason why the HC measurement was included with the best available technology limited by size, weight, power (need of a heated exhaust line) and safety (HC-FID hydrogen bottles on a motorcycle on road are not allowed). This is stated in page 3, line 8. **

Figure 8: this figure shows only results from MiniPEMS #2. What about Mini PEMS #1? Is this paper evaluating the one, two, or both?

** Both are evaluated. Figure 8 caption amended: "Relative deviations between PEMS (PEMS no. 1 unless otherwise noted) and laboratory emission factors." **

-Supplementary, Figure S4: It is not clear what is represented in these two graphics. Is it for tests 1 and 2? I suggest identifying them, even because in Table S6 is described three tests

** Caption amended: "Two different R47 tests shown (summary statistics in Table S8)." **

Technical corrections: - I don't understand why tables and figures are at the end of the paper. If it is an editor requirement, ok; if not, it will be easier to follow the authors' thought if figures and tables are inserted into the text.

** We agree with the Reviewer. Unfortunately we realized late that only during final submission tables and figure should go at the end for editorial requirements. We are sorry if this caused difficulty in reading the manuscript. As this is the last step of the

review process we decided to keep this format. **

- Equations 1, 2, 3 and 4: poor resolution; it is missed a space (white line) before and after equation 4 -

** All Equations have been rewritten. **

Page 12, line31: Text makes reference to Table S8 but I believe that the correct is Table S6 (also forpage 14, line 28)

** Table numbering in Supplemental info corrected (Table S8 is now the last table, the reference in the text is correct). **

- Page 27, Table 6: It would be interesting to show three significant algharisms instead of only two for all pollutants except CO2, e.g., HC on the bench, test 1, 0.209 instead of 0.21

** Amended as suggested. **